# Examining the Impact of Agency Issues on Corporate Performance: A Bibliometric Analysis

**Vinay Khandelwal** [1,2,*], **Prasoon Tripathi** [3], **Varun Chotia** [4], **Mohit Srivastava** [5], **Prashant Sharma** [6] **and Sushil Kalyani** [7]

1. Institute of Business Management, GLA University, Mathura 281406, India
2. Stirling Management School, University of Stirling, Stirling FK9 4LA, UK
3. Institute of Management Studies, Ghaziabad 201015, India; prasoonmtripathi@gmail.com
4. Jaipuria Institute of Management Jaipur, Jaipur 302033, India; varun.chotia@jaipuria.ac.in
5. Department of Strategy and Entrepreneurship, EM Normandie Business School, 76600 Metis Lab, France
6. Jindal School of Banking & Finance (JSBF), O. P. Jindal Global University, Sonipat 131001, India; prashant.sharma@jgu.edu.in
7. Department of Management, NIIT University, Neemrana 301705, India; sushilk@niituniversity.in
* Correspondence: vinay.khandelwal@gla.ac.in

**Abstract:** An agency problem is defined as a conflict of interest arising due to a misalignment of interests among the managers and other stakeholders of the company. This article aims to review the articles addressing the agency problem and their impact on business performance. This article reviews the contributions of prominent theorists on agency problems and agency costs. Using bibliometric attributes of 740 articles from the Scopus database, this study highlights the publishing trend and outlets, along with leading contributors and collaborators in terms of authors, institutions, and countries. This study identifies the clusters through the bibliographic coupling technique and a trend topics analysis. Most researchers have focused on corporate governance and expressed the agency problem as one of the impact areas. This study is unique as no study to date specifically focuses solely on agency theory or the agency problem through the lens of bibliometric analysis. Future research directions on agency problems and their solutions conclude this study.

**Keywords:** agency problem; agency cost; agency theory; bibliometric; business performance; corporate governance

## 1. Introduction

The principal–agent problem (or the agency dilemma) occurs when one entity (the "agent") is employed to make decisions and/or take actions on behalf of, or impacts, another entity (the "principal"). The dilemma happens when agents act in their best interests, contrary to principals' interests. This problem usually arises when both entities maximize their interests. When agents focus on their own gains before the principal's gains, it is called an agency problem (AP). The emergence of agency theory and the associated problems is rooted in the complexities arising from the separation of ownership and control within organizations. Originating in the 1970s, agency theory became a pivotal framework employed across diverse disciplines such as economics, law, finance, accounting, and political science. Initially introduced by Jensen and Meckling (1976), the theory gained traction due to its applicability in analyzing the challenges arising when one entity, the agent, acts on behalf of another entity, the principal, often leading to misalignments of interests. Ownership separation from control in big companies leads to a conflict of interests among shareholders and management. The firm's managers often focus on personal goals that conflict with the shareholders' wealth maximization objective (Shaifali 2019). The issues that arise among principals and their agents are often due to a lack of congruence in their approach because of information asymmetry (Jiang 2023). Information asymmetry

happens when either party has more information than the other. Thus, the main focus of both principals and agents should be on resolving APs and saving on agency costs. Panda and Leepsa (2017) define agency costs as the internal costs arising from the misalignment of interests of the agent and the principal. It constitutes the cost of selecting and recruiting a suitable agent, costs incurred in setting benchmarks, overlooking the agent's actions, the bonding costs, and the residual loss arising from conflicts between the management and shareholders. Scholars researching Agency Theory (AT) study the relationship between principals and agents, and suggest ways to minimize the occurrences of agency issues and, ultimately, agency costs.

The principal and agent theory emerged in the 1970s from the combined economics and institutional theory disciplines. The theory was taken up by researchers in several disciplines, like strategy (Barnard 1938), law (Banfield 1985), economics and finance (Jensen and Meckling 1976), accounting (Baiman 1990), and political science (Mitnick 1982), among many. Researchers use agency theory to analyze the top leaders in big private and public enterprises. Given its roots in economics, agency theory suggests that the agents who work in an organization have a utility maximization logic and seek to get what is in their best interest, even when it is not in the best interest of the organization (Eisenhardt 1989). Based on the essential contributions of the work of Barnard (1938) on cooperation in organizations, agency theory focuses on the conflict between objectives, created by various individuals who, while engaged in these organizations, seek what is in their best interest.

The number of bibliometric studies on AP is limited (Bendickson et al. 2016). Past studies have focused a lot on corporate governance (Jahja et al. 2020; Naciti et al. 2022), boards of directors (Pascual-Fuster and Crespí-Cladera 2018), or more specific topics, such as board diversity and its impact on CSR (Baker et al. 2020a; Do 2023; Eliwa et al. 2023). This study differs from other published reviews because, to the best of the authors' knowledge, this research is the first bibliometric study that focuses primarily on the agency problem (AP) and its impact on financial performance across business fields with language, scholarly, and subject filtration in the Scopus database. This review focuses on mapping the domain of AP research through a bibliometric analysis. The insights on the current scenario and future research directions are shared after different analyses on AP. Thus, this study has the below-mentioned Research Questions (RQs):

- What is the trend in publications on AP?
- Which are the most influential publishing outlets for research on AP?
- Who are the prolific contributors to the field of AP?
- What are the themes and clusters for research on AP?
- What are the future research areas in the field of AP?

The remaining sections of this document are arranged as follows. The second section discusses the background of AP. The third section describes the methodology applied for this study. Results and discussions for all analyses are summarized in the fourth section. Further sections contain the research themes and future research directions to strengthen the field of AP.

## 2. Theoretical Background

Though the problem of the agency has existed for a very long time, Smith was the first author to ever write about it (Seth 2018). He forecasted that if the management of an organization is handed over to a person or a group of persons other than the owners, then it is likely that they may not work for the benefit of the owners. Bhabra and Wood (2014) discussed the ownership structure of large firms operating in the USA and argued that agents may use the assets of the organization to maximize their interests. The roots of agency theory trace back to seminal works that have shaped its conceptual foundation. Berle and Means' groundbreaking work in 1932, particularly in "The Modern Corporation and Private Property", laid the groundwork for understanding the challenges arising from the separation of ownership and control in large corporations. Moving forward, Eisenhardt's influential theories significantly advanced the discourse by addressing the intricacies

of control mechanisms within organizations (Eisenhardt 1985, 1989). These milestones underscore the theoretical evolution of agency theory, emphasizing shifts in focus from corporate governance dynamics to nuanced examinations of principal–agent relationships. Furthermore, pivotal contributions by scholars such as Jensen and Meckling in their 1976 paper, "Theory of the Firm: Managerial Behavior, Agency Costs, and Ownership Structure", have been instrumental in defining the theoretical landscape of agency issues.

Jensen and Meckling (1976) discussed three types of agency costs—monitoring costs, bonding costs, and residual losses. Monitoring costs are incurred by the principal to oversee the conduct and limit the aberrant activities of its agent. Bonding expenses are incurred to ensure that agents do not make certain decisions that may impact the principal's interests. The residual losses arise due to the misalignment of interests of the principal and the agent and are measured in terms of the dollar equivalent of the losses to the principal. Often the agents tend to underdeliver on their promises to the principal to maximize their gains. This is referred to as a 'moral hazard'. Also, the more autonomy an agent gets to conduct complex work, the more significant the moral hazard becomes (Cowden et al. 2020). As per theorists, there are two main reasons behind principal–agent problems—one arising out of different risk preferences of the principals and the agents, and another arising since both the principals and agents are rational human beings and work towards maximization of their self-interests. Managers may misbehave if their interests differ from those of the company (Dalton et al. 2007).

Panda and Leepsa (2017) segregated the AP into three types. The first type occurs amongst the principal and agents, due to the different levels of risk appetite, information asymmetry, and self-satisfying behavior based on the rational behavior of human beings (Elfenbein and Knott 2015), which states that rational individuals maximize their interests. This misalignment in interests of agents and the principals gives rise to the principal–agent problem. The second type of AP happens between the major and minor shareholders in a company. Shareholders with major holdings have a higher weight in voting and are likely to make decisions for their benefit which may obstruct the interests of shareholders with a lesser stake in the company. This problem is usually found in companies with higher ownership proportion gaps (Fama and Jensen 1983). The third type of problem arises because of risk preferences between the principals and creditors of the company. Quite often, projects are funded with more debt and less equity, as financing completely through equity is expensive (Jiraporn et al. 2012; Khandelwal et al. 2023; Narayan et al. 2021). Some projects are subject to a high risk of default. If such a project is successful, good premiums are enjoyed by the shareholders, and creditors are paid at a pre-decided rate of interest; however, if the project is unsuccessful, the creditors are asked to accept partial settlements due to loss in projects. This problem is seen in companies engaging in project financing. This leads to creditors being stuck with lesser returns for high risks.

## 3. Review Methodology

### 3.1. Search Strategy

The search strategy for this review meticulously employed a three-step process to ensure the inclusion of relevant articles while adhering to specific criteria. The first step involved a database search, primarily focusing on the Scopus database due to its comprehensive coverage and reliable bibliometric parameters (Archambault et al. 2009; Kumar et al. 2021; Mongeon and Paul-Hus 2016). The search targeted articles related to agency theory using the keywords "agency cost" and "agency problem" with the 'OR' operator, forming the foundational elements of agency theory. To narrow down the focus, additional keywords like "performance*" and "profit*" were included with the Boolean operator '*' to capture all keywords starting with "profit" (Tripathi et al. 2023). The study specifically concentrated on articles related to 'business', 'organization', or 'firm'. Exclusion criteria were then applied, excluding articles from 2022 and limiting the search horizon to 2021. The second step involved subject filtration, considering only articles within the "Business, Management, and Accounting" category in the Scopus database, aligning with the overarching

discipline where agency theory resides. The third and final step incorporated scholarly filtration, restricting the review to research articles published in English, thereby excluding other languages and publication types such as conference proceedings, reviews, books, and book chapters (Mukherjee et al. 2022). Through this comprehensive inclusion and exclusion criteria framework, the study ultimately reviewed a total of 740 documents, ensuring a focused and relevant dataset for analysis.

*3.2. Bibliometric Analysis*

Bibliometric analysis serves as an invaluable methodological tool in scrutinizing the state of research within complex domains such as AP (Naciti et al. 2022). Its utility lies in its ability to systematically evaluate and quantify the existing body of literature, offering insights into the trends, contributors, and thematic clusters shaping the field (Mukherjee et al. 2022). By employing bibliometric analysis, this study navigates the expansive landscape of AP research, unraveling patterns that might be challenging to discern through traditional literature reviews. This study applied a comprehensive bibliometric analysis to examine 740 selected publications on AP. Extracting bibliometric data from the Scopus database, an array of analyses explored the landscape of AP research. The investigation covered publication years, authors, journal titles, citations, institutes, and countries, addressing specific research questions. The study aimed to discern evolving publishing trends (RQ1), identify prominent outlets in the field (RQ2), and highlight top-performing authors, institutions, and countries (RQ3). The bibliometric analysis also delved into keyword exploration through co-occurrence analysis, forming knowledge clusters that delineate sub-themes within the AP domain (RQ4). Inspired by Donthu et al. (2021), this approach assessed the impact and centrality of each knowledge cluster. Within these clusters, articles were scrutinized to ascertain current research topics (RQ5) and identify gaps in the existing literature, shaping the future research agenda.

To implement the bibliometric analysis, the study utilized the bibliometrix package in the R software (version 4.3.1) environment, facilitated by the RStudio platform. Specifically, the 'Biblioshiny' command harnessed bibliometric techniques, including the identification of top authors, sources, and articles, as well as the analysis of countries, institutions, and trending keywords. Additionally, science mapping was employed to visually represent knowledge clusters, providing a comprehensive overview of interconnections and focal points within the AP research landscape. Through this multifaceted approach, the study aims to contribute a nuanced understanding of the current state and future directions of research on AP and its implications for corporate performance.

## 4. Results and Discussion

The study highlights that the earliest articles on AP were published in 1985, and the total research articles indexed in Scopus till 2021 stand at 740 after the language, scholarly, and subject filtration. This section further contains detailed findings on the bibliometric attributes of the articles under study. Firstly, the line chart represents the year-wise publications corresponding to the year of publication (Figure 1). Secondly, the top publishing outlets are listed in order of decreasing total citations (Table 1, Figure 2).

**Table 1.** Top Performing Publishing Outlets in the Research Domain of Agency Theory.

| Publishing Outlet | h-Index | TC | NP | PY-Start |
|---|---|---|---|---|
| Strategic Management Journal | 17 | 2681 | 19 | 1991 |
| Academy Of Management Journal | 13 | 2676 | 13 | 1996 |
| Journal Of Financial Economics | 9 | 1600 | 9 | 1995 |
| Management Science | 8 | 1503 | 10 | 1985 |
| Journal Of Management | 11 | 1353 | 12 | 2001 |
| Journal Of Accounting And Economics | 8 | 1087 | 10 | 1987 |
| Journal Of Corporate Finance | 14 | 1079 | 19 | 1996 |
| Corporate Governance: An International Review | 14 | 756 | 24 | 1994 |

**Table 1.** *Cont.*

| Publishing Outlet | h-Index | TC | NP | PY-Start |
|---|---|---|---|---|
| Accounting Review | 5 | 698 | 7 | 1997 |
| Journal Of Finance | 3 | 583 | 3 | 2004 |
| Journal Of Business Ethics | 11 | 527 | 16 | 1991 |
| Journal Of International Business Studies | 2 | 483 | 2 | 2004 |
| Journal Of Business Research | 5 | 413 | 5 | 2005 |
| The Journal Of Finance | 1 | 377 | 1 | 1993 |
| Academy Of Management Review | 1 | 369 | 1 | 2005 |
| Review Of Finance | 3 | 340 | 3 | 2011 |
| Journal Of Financial And Quantitative Analysis | 3 | 328 | 3 | 1987 |
| Review Of Financial Studies | 3 | 310 | 3 | 2012 |
| Marketing Science | 8 | 302 | 8 | 1997 |
| Harvard Law Review | 1 | 300 | 1 | 2004 |

Note: Articles are Ranked based on total citations received, TC—Total Citations, NP—Number of Publications, PY-Start—Publication Year Start.

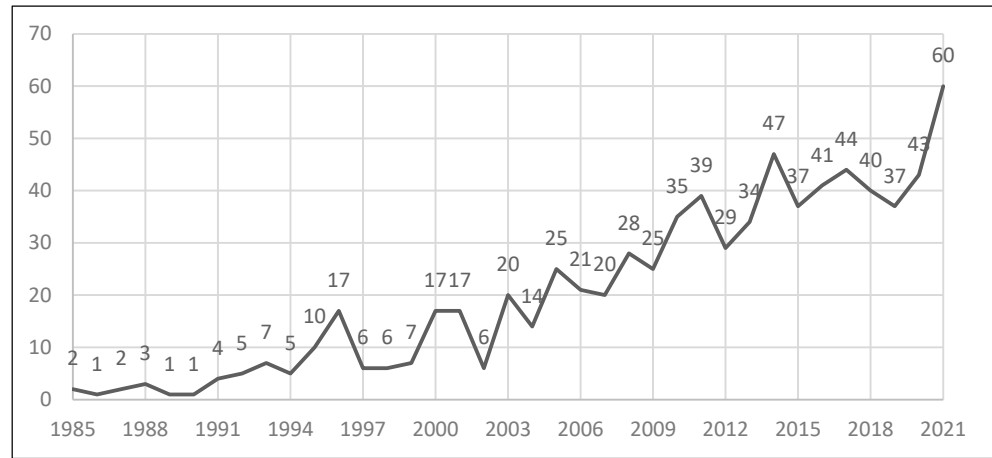

**Figure 1.** Publishing Trend of Research on Agency Theory.

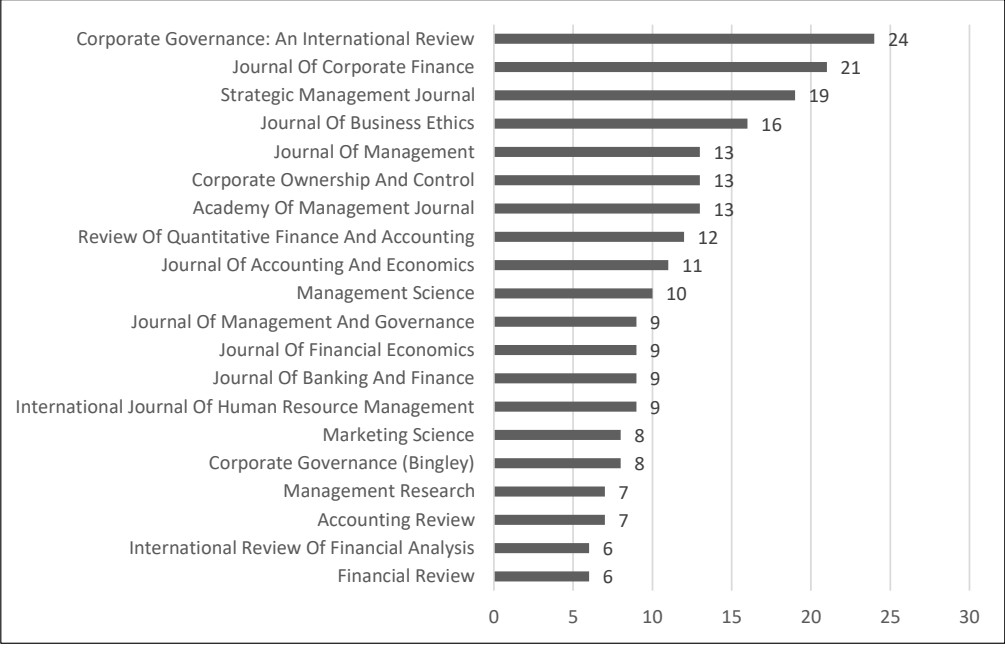

**Figure 2.** Leading Publishing Outlets of Research on Agency Theory (Minimum of Six Articles).

### 4.1. Publishing Trend

The line plot depicts the published articles of each year following the search strategy. As evident from Figure 3, it is evident that AP has seen increased scholarly participation over the previous 36 years. The highest number of articles were published in 2021 (n = 60), being the most recent year of the study. A sharp growth is observed from 2002 with increased outputs each year hence.

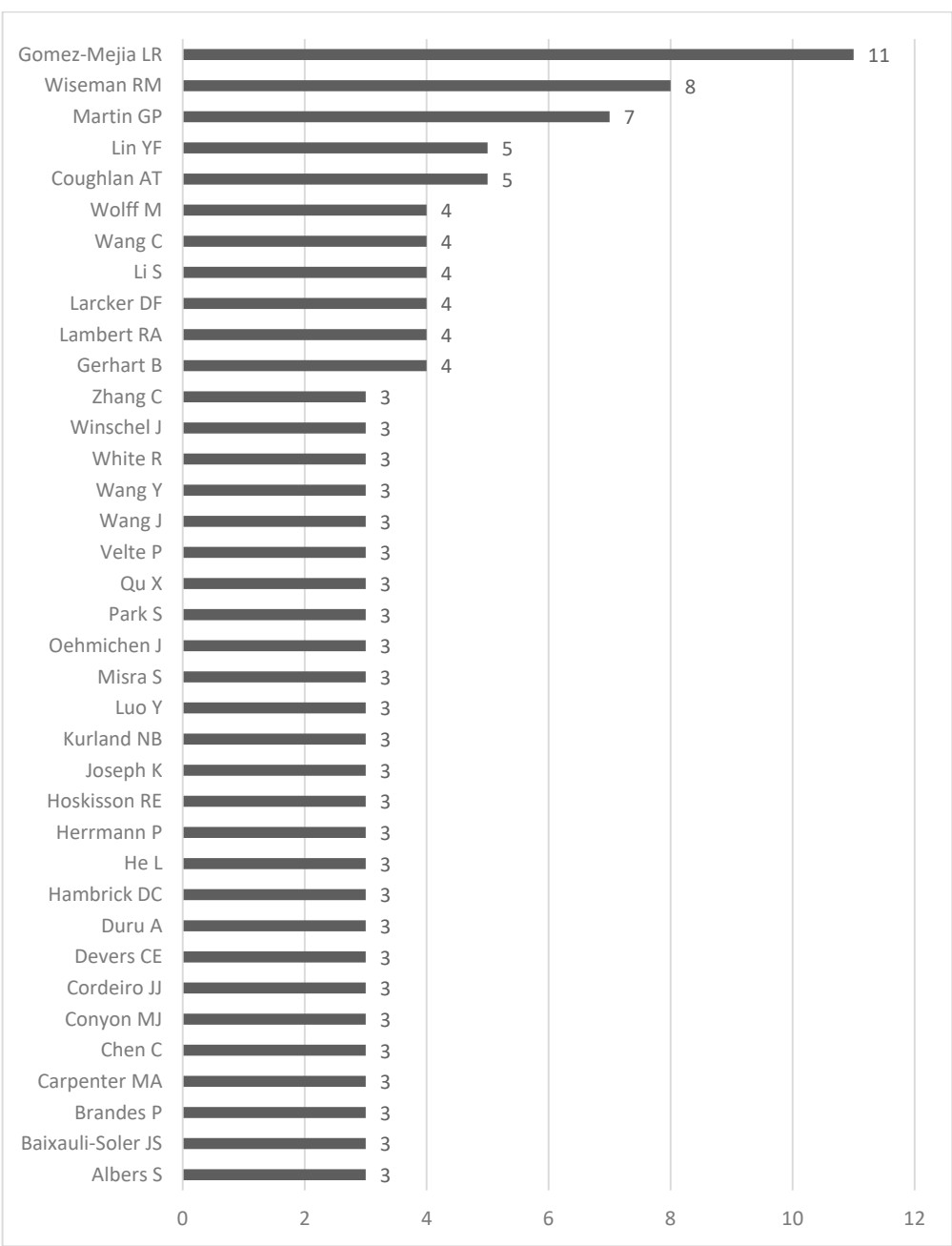

**Figure 3.** Leading Authors Contributing to Research on Agency Theory (Minimum of Three Research Articles).

### 4.2. Publishing Outlets

The analysis of documents by publishing outlets reveals the top journals publishing articles on AP. This study lists the top 20 journals, sorted in order of total citations on articles, in Table 1. The journals are listed with their corresponding h-index, total citations on articles on AP, the number of publications, and the publication year start.

The study puts the 'Strategic Management Journal' by Wiley at the first rank based on total citations. Interestingly, the 'Strategic Management Journal' is also the best journal based on its h-index for the study. This is followed by 'Academy of Management Journal' by the Academy of Management, and 'The Journal of Financial Economics' by Elsevier. Figure 4 depicts the top 20 publishing outlets based on the number of papers contributed to the existing literature on AP. The 'Corporate Governance: An International Review' is the highest contributor in this field, followed by the 'Journal of Corporate Finance' and 'Strategic Management Journal', respectively.

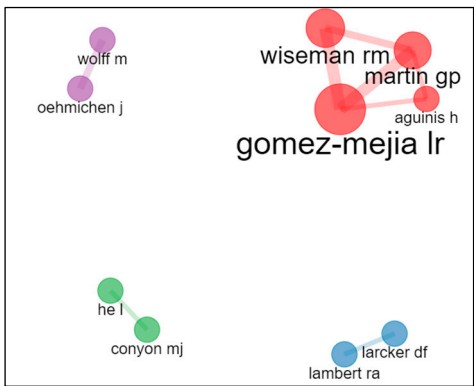

**Figure 4.** Prominent Collaborators (Authors) on the Research Topic of Agency Theory.

*4.3. Publication Performance*

4.3.1. Global Citations

Global citations refer to the count of all articles globally that have cited the study without any filtration (e.g., language, scholarly, subject, etc.) (Baker et al. 2020b). Table 2 summarizes the articles on AP in decreasing order of their total global citations. In this study, we find that the article with the most global citations is "Control: Organizational and Economic Approaches", published in 1985 in the journal "Management Science". It is cited a total of 1227 times globally, and is followed by the article titled "Internationalization and firm governance: The roles of CEO compensation, top team composition, and board structure", published in 1998 in the "Academy of Management Journal" with a citation count of 613.

**Table 2.** Leading 10 Articles in Research Domain of Agency Theory, based on Total Global Citations.

| Article | Authors and Year | Journal | TGC |
|---|---|---|---|
| Control: Organizational and Economic Approaches | (Eisenhardt 1985) | Management Science | 1227 |
| Internationalization and firm governance: The roles of CEO compensation, top team composition, and board structure | (Sanders and Carpenter 1998) | Academy of Management Journal | 613 |
| Do corporations award CEO stock options effectively? | (Yermack 1995) | Journal of Financial Economics | 524 |
| Managing foreign subsidiaries: Agents of headquarters, or an interdependent network? | (O'Donnell 2000) | Strategic Management Journal | 473 |
| The choice of performance measures in annual bonus contracts | (Ittner et al. 1997) | Accounting Review | 466 |
| Contracting theory and accounting | (Lambert 2001) | Journal of Accounting and Economics | 437 |
| Board control and CEO compensation | (Boyd 1994) | Strategic Management Journal | 432 |
| Why do corporate managers misstate financial statements? The role of option compensation and other factors | (Efendi et al. 2007) | Journal of Financial Economics | 425 |
| Managing knowledge transfer in MNCs: The impact of headquarters control mechanisms | (Björkman et al. 2004) | Journal of International Business Studies | 415 |
| Top-Management Compensation and Capital Structure | (John and John 1993) | The Journal of Finance | 377 |

Note: TGC—Total Global Citations.

4.3.2. Local Citations

Local citations refer to the count of all articles in the review corpus that have cited the study (Mukherjee et al. 2022). Alternatively, local citations are the citations received

on the article from the current study sample of 767 articles after the language, scholarly, and subject filtration of the Scopus database. From this study, we find that the article titled "Board control and CEO compensation" published in Strategic Management Journal in 1994 has been cited by 49 articles (6.4%). This is followed by the article titled "Do corporations award CEO stock options effectively?", published in the Journal of Financial Economics in 1995 with a local citation count of 45 articles (5.9%). The article ranking based on local citations is summarized in Table 3.

**Table 3.** Leading 10 Articles in the Research Domain of Agency Theory based on Total Local Citations.

| Article Title | Authors and Year | Journal | TLC |
|---|---|---|---|
| Board control and CEO compensation | (Boyd 1994) | Strategic Management Journal | 49 |
| Do corporations award CEO stock options effectively? | (Yermack 1995) | Journal of Financial Economics | 45 |
| Executive compensation: A multidisciplinary review of recent developments | (Devers et al. 2007) | Journal of Management | 37 |
| Top-Management Compensation and Capital Structure | (John and John 1993) | The Journal of Finance | 36 |
| Control: Organizational and Economic Approaches | (Eisenhardt 1985) | Management Science | 26 |
| Executive compensation and corporate governance in China | (Conyon and He 2011) | Journal of Corporate Finance | 22 |
| Moving closer to the action: Examining compensation design effects on firm risk | (Devers et al. 2008) | Organization Science | 22 |
| An empirical investigation of the role of subjective performance assessments versus objective performance indicators as determinants of CEO compensation | (Caranikas-Walker et al. 2008) | Management Research | 22 |
| Is CEO pay in high-technology firms related to innovation? | (Balkin et al. 2000) | Academy of Management Journal | 22 |
| The choice of performance measures in annual bonus contracts | (Ittner et al. 1997) | Accounting Review | 22 |

Note: TLC—Total Local Citations.

### 4.4. Prolific Authors and Collaborations

#### 4.4.1. Prolific Authors

The analysis of literature on AP reveals that Kathleen M. Eisenhardt, a professor in the school of engineering at Stanford University, has the highest number of total citations with a count of 1227 citations of articles on agency problems. Her first publication was in the year 1985 entitled "Control: Organizational and economic approaches" wherein she discussed agency theory and control (Eisenhardt 1985). She is followed by the late Mason A. Carpenter of the University of Wisconsin-Madison with 851 total citations in the field. The ranking of authors based on the number of papers published is shown in Table 4. As per the number of documents, Luis Gomez-Mejia of Arizona State University has the highest published on AT ($n_p$ = 11). He is followed by Robert M Wiseman of Michigan State University with eight published articles on AP (see Figure 5).

**Table 4.** Top Performing Authors in the Research Domain of Agency Theory.

| Authors | h-Index | TC | NP | PY-Start |
|---|---|---|---|---|
| Eisenhardt KM | 1 | 1227 | 1 | 1985 |
| Carpenter MA | 3 | 851 | 3 | 1998 |
| Gomez-Mejia LR | 10 | 765 | 11 | 2000 |
| Lambert RA | 4 | 701 | 4 | 1991 |
| Hambrick DC | 3 | 656 | 3 | 1995 |
| Sanders WG | 1 | 613 | 1 | 1998 |
| Larcker DF | 4 | 581 | 4 | 1991 |
| Finkelstein S | 2 | 558 | 2 | 1995 |
| Wiseman Rm | 6 | 542 | 8 | 2002 |
| Devers CE | 3 | 525 | 3 | 2006 |
| Yermack D | 1 | 524 | 1 | 1995 |
| Ittner CD | 2 | 486 | 2 | 1997 |
| O'Donnell SW | 1 | 473 | 1 | 2000 |
| Rajan MV | 1 | 466 | 1 | 1997 |

**Table 4.** *Cont.*

| Authors | h-Index | TC | NP | PY-Start |
|---|---|---|---|---|
| Boyd BK | 2 | 465 | 2 | 1994 |
| Conyon | 3 | 462 | 3 | 2006 |
| Bjrkman I | 2 | 431 | 2 | 2000 |
| Efendi J | 1 | 425 | 1 | 2007 |
| Srivastava A | 1 | 425 | 1 | 2007 |
| Swanson EP | 1 | 425 | 1 | 2007 |

Note: Authors are ranked on the basis of total citations received, TC—Total Citations, NP—Number of Publications, PY-Start—Publication Year Start.

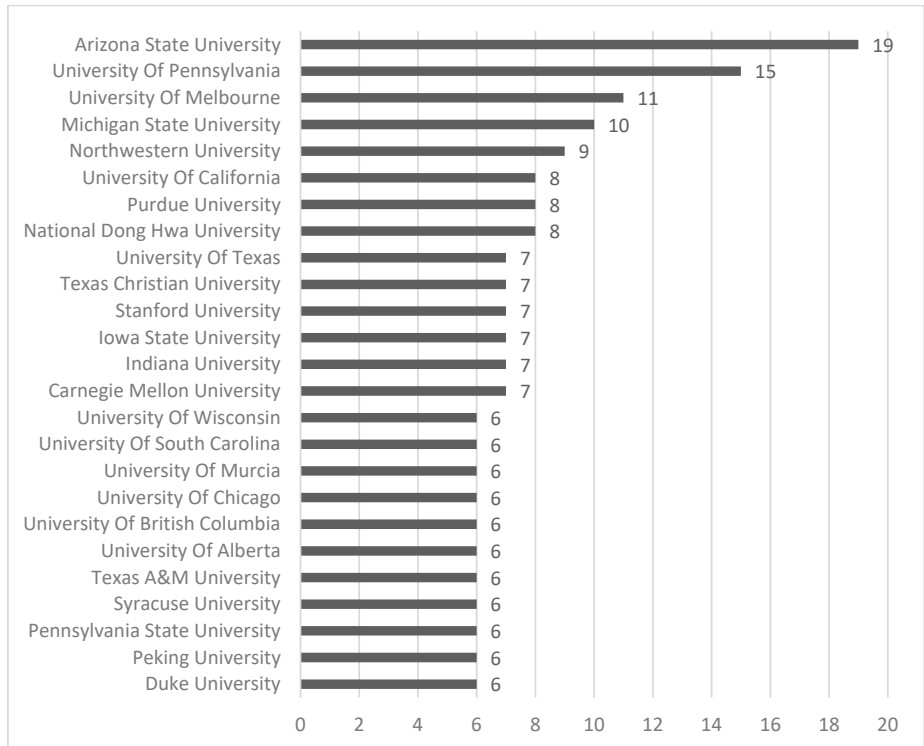

**Figure 5.** Leading Institutions Contributing to Research on AP (Minimum of Six Research Articles).

### 4.4.2. Author Collaborations

The co-authorship analysis reveals the nature and groups of authors, which is similar to the social network of researchers working on a common project (Donthu et al. 2021). van Eck and Waltman (2010) stated that the co-authorship networks have been studied extensively; however, the visualization of such networks has been given little attention. The analysis reveals the prominent collaborative groups in Figure 6 on the research topic of AP. The collaborative author group is Luis Gomez-Mejia of Arizona State University, Robert M Wiseman of Michigan State University, Geoffrey Martin of Melbourne Business School, and Herman Aguinis of George Washington University. The size of the circles in Figure 4 resembles the influence of the author. Other collaborative groups include Michael Wolff and Jana Oehmichen of the University of Groningen, Lerong He of State University of New York at Brockport and Martin Conyon of Bentley University, and, lastly in the figure, David F. Larcker of Stanford University and Richard A Lambert of Northwestern University.



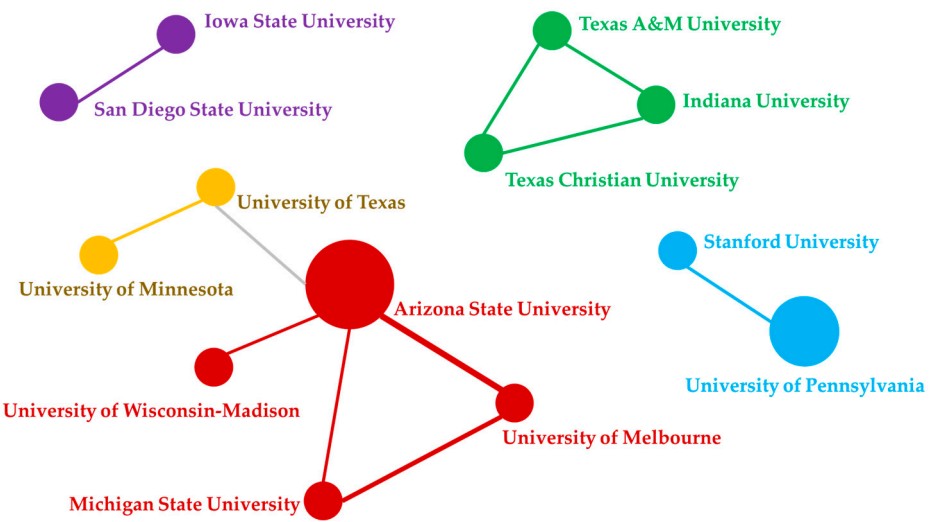

**Figure 6.** Prominent Collaborators (Institutions) on the Research Topic of Agency Theory.

*4.5. Countries*

4.5.1. Contributing Countries

Figure 7 summarizes the country-wise academic contributions in the field of AP. Notably, Alaska and United States dominate the field with more than 800 citations in the countries. They are followed by China with cited research in the bracket of 200–400 studies. Other countries that are highlighted in red represent lower impact output within the range of 0–200 citations. The nations that are shown in white color denote no or very little involvement in the academic research in this area.

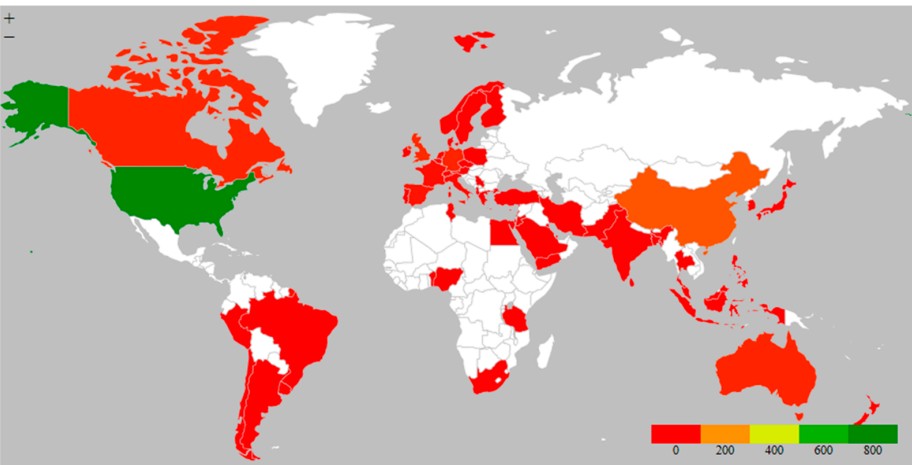

**Figure 7.** Geographic Heat Map of Countries Contributing to Research on Agency Theory.

4.5.2. Country Collaborations

This study visualizes the country-wise collaborations in the form of a network diagram (Figure 8). Cardoso et al. (2020) presented a country research performance model to evaluate a country's research dominance. They considered the countries' overall performance, the countries' journals' performance, and the countries' institutions' performance to ascertain the countries' research dominance. However, the researchers missed out on studying the cross-country collaborations on the topic. In Figure 8, the country-wise collaboration network is depicted. The countries marked with identical colors are part of the same cluster. The countries in the same clusters are shown to work together over the countries marked with different colors. Five country-wise clusters can be observed from the figure, with the biggest cluster dominated by the USA, China, Canada, and Australia. The second cluster

consists of European countries such as the United Kingdom, Italy, Poland, and Cyprus, along with Pakistan. The third cluster in terms of its size is France, Finland, Norway, and Tunisia. This is followed by the fourth cluster of Belgium, Germany, and Netherlands, and, lastly, a separate and unrelated cluster of Indonesia and Malaysia are shown to have researched together on AP.

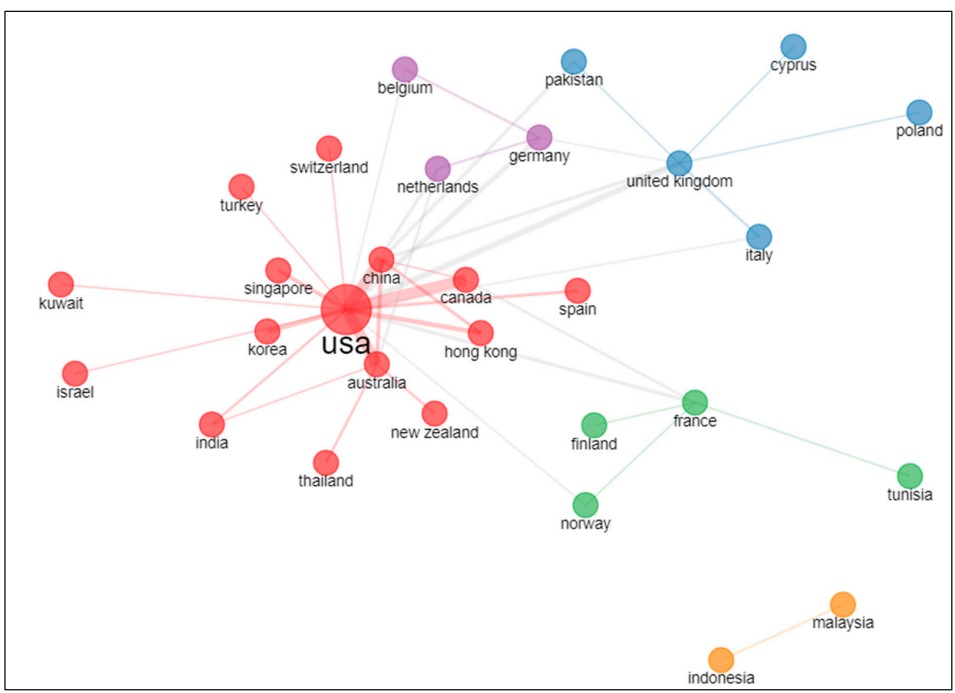

**Figure 8.** Prominent Collaborators (Countries) on the Research Topic of Agency Theory.

*4.6. Institutions*

4.6.1. Leading Institutions

The analysis indicates that Arizona State University made the highest contribution to the field of AP in the past with 19 articles published (see Figure 9). This is followed by the University of Pennsylvania with 15 articles. These are followed by the University of Melbourne, Michigan State University, and Northwestern University with eleven, ten, and nine articles respectively. The highest contribution of Arizona State University can be credited to Prof. Luis Gomez-Mejia who is also the leading author in the field. Contrastingly, the contribution of Robert M Wiseman and Richard A Lambert is also significant for boosting the impact of Michigan State University and Northwestern University.

4.6.2. Institutional Collaborations

The network visualization diagram of institutional co-authorship reveals five major collaboration groups (see Figure 10). The biggest group consists of Arizona State University, the University of Melbourne, Michigan State University, and the University of Wisconsin-Madison. The next group consists of Texas A&M University, Indiana University, and Texas Christian University. The two groups are followed by collaborative duos of Stanford University–University of Pennsylvania, Iowa State University–San Diego State University, and University of Texas–University of Minnesota.

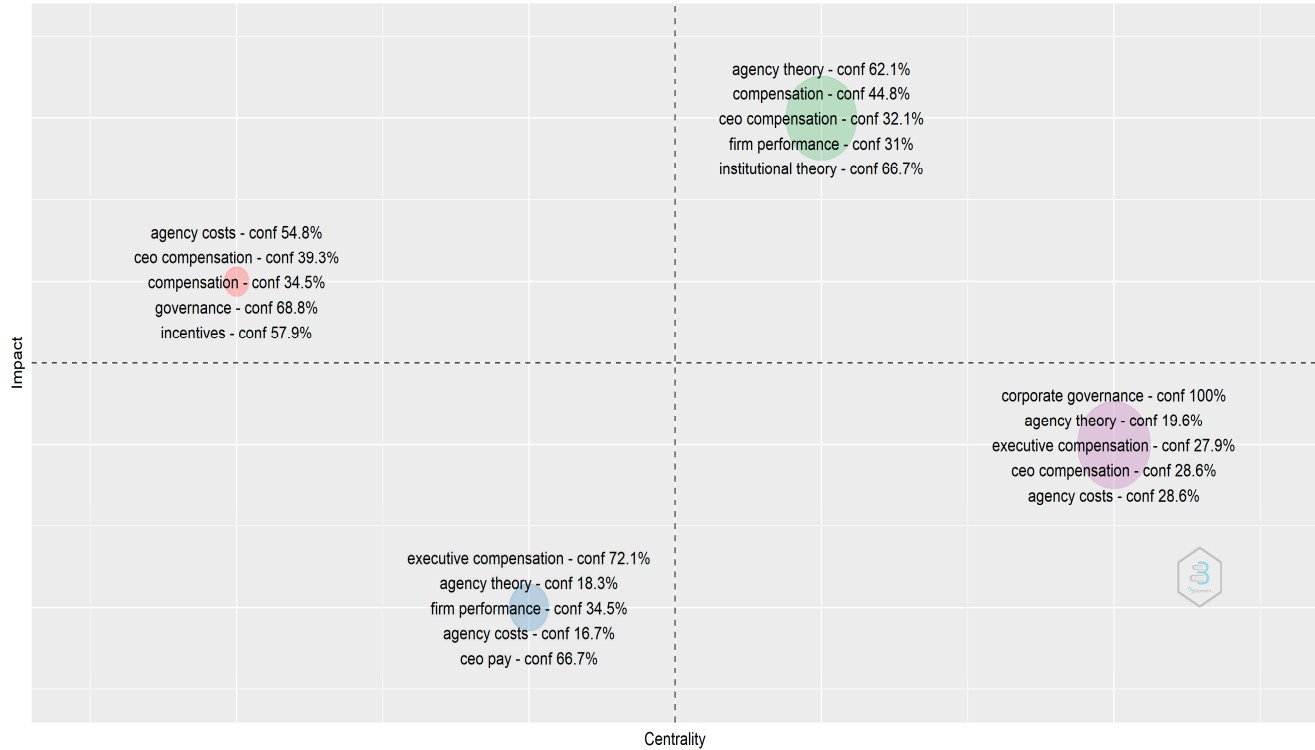

**Figure 9.** Knowledge Clusters Identified as a Result of "Bibliographic by Coupling" Analysis.

*4.7. Themes*

The bibliographic coupling of documents is used to form knowledge clusters. The knowledge clusters contain documents of similar themes underlining the thematic structure of AP. Following the methodology shared by Kumar et al. (2021), we form knowledge clusters by the bibliographic coupling of documents based on authors' keywords. Four knowledge clusters are formed from the analysis; Table 5 summarizes the keywords and their occurrence for each cluster. The knowledge clusters are also plotted in Figure 9 based on their centrality and impact. Centrality in bibliometric research refers to the prominence of a publication or author within a scholarly network, often measured by the number and strength of connections. Impact of the cluster assesses the influence and significance of a research output, typically measured by citations and other indicators of scholarly impact (Sahoo et al. 2023). Impact measures the extent to which a research output is cited and acknowledged within the context of the discussed theme, highlighting its influence and relevance in the scholarly discourse (Sahoo et al. 2022). Table 6 lists the top ten most relevant documents for each cluster.

**Table 5.** Descriptive Summary of Formed Knowledge Clusters.

| # | Knowledge Cluster | Keyword (% of Occurrences) | Frequency | Centrality | Impact |
|---|---|---|---|---|---|
| 1 | Corporate Governance | Agency costs (54.8%), CEO compensation (39.3%), compensation (34.5%), governance (68.8%), and incentives (57.9%) | 90 | 0.08 | 2.58 |
| 2 | Agency Costs and Governance | Executive compensation (72.1%), agency theory (18.3%), firm performance (34.5%), agency costs (16.7%), and CEO pay (66.7%) | 106 | 0.21 | 2.42 |
| 3 | Agency Theory and Compensation | Agency theory (62.1%), compensation (44.8%), CEO compensation (32.1%), firm performance (31%), and institutional theory (66.7%) | 150 | 0.21 | 2.70 |
| 4 | Executive Compensation and Agency Costs | Corporate governance (100%), agency theory (19.6%), executive compensation (27.9%), CEO compensation (28.6%), and agency costs (28.6%) | 154 | 0.24 | 2.44 |

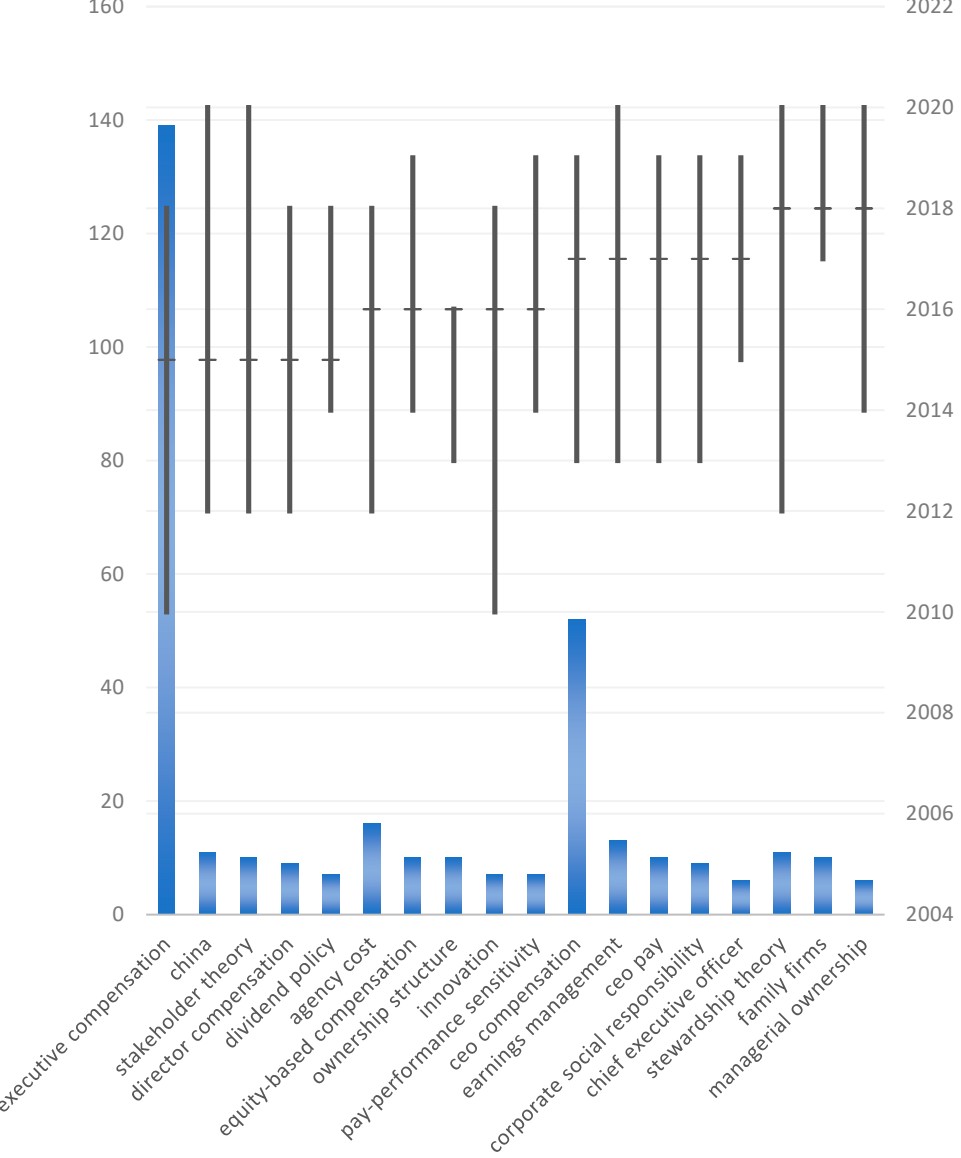

**Figure 10.** In-trend keywords on the research in the field of AT over the past seven years.

The first cluster comprises research articles on Corporate Governance. Westphal and Zajac (1995) suggested that high incentives and monitoring costs are not optimal. A firm's strategy should focus on corporate governance implications equally as product and market implications. Devers et al. (2007, 2008) in their studies on executive compensation factors and their robustness, revealed that no theoretical model is strong enough to determine optimal executive compensation. van Essen et al. (2012) find that strong directors can establish tighter links between executive pay and firm performance. Scholarship has further highlighted that CEOs that have higher board influence enjoy higher compensation packages and only shareholders and their agents can control them. The articles in this cluster are highest in terms of centrality to the theme of AP and have a high impact on the literature.

**Table 6.** Most relevant documents cluster-wise sorted on normalized local citation score.

| Cluster | Documents | Cluster Size | Normalized Local Citation Score |
|---|---|---|---|
| Cluster 1: Executive Compensation and Governance | | | |
| 1 | (Brockman et al. 2010) | 90 | 5.76 |
| 1 | (Bolton et al. 2015) | 90 | 2.18 |
| 1 | (Gillan et al. 2021) | 90 | 1.84 |
| 1 | (Dhole et al. 2016) | 90 | 1.78 |
| 1 | (Ertugrul and Hegde 2008) | 90 | 1.33 |
| 1 | (Masulis and Reza 2015) | 90 | 1.09 |
| 1 | (Eisdorfer et al. 2015) | 90 | 1.09 |
| 1 | (Berry and Junkus 2013) | 90 | 1.05 |
| 1 | (Imes and Anderson 2021) | 90 | 1 |
| 1 | (Khandelwal and Chotia 2022) | 90 | 1 |
| Cluster 2: CEO Compensation | | | |
| 2 | (Conyon and He 2011) | 106 | 9.2 |
| 2 | (van Essen et al. 2012) | 106 | 7.62 |
| 2 | (Andreas et al. 2012) | 106 | 5.58 |
| 2 | (Michiels et al. 2013) | 106 | 4.25 |
| 2 | (Hong et al. 2016) | 106 | 3.57 |
| 2 | (Callan and Thomas 2014) | 106 | 3.36 |
| 2 | (Smirnova and Zavertiaeva 2017) | 106 | 3.32 |
| 2 | (O'Reilly and Main 2010) | 106 | 2.88 |
| 2 | (Sun and Shin 2014) | 106 | 2.52 |
| 2 | (Sheikh et al. 2019) | 106 | 2.4 |
| Cluster 3: Agency Theory | | | |
| 3 | (Nyberg et al. 2010) | 150 | 5.35 |
| 3 | (Francoeur et al. 2021) | 150 | 3.32 |
| 3 | (Coles et al. 2001) | 150 | 2.96 |
| 3 | (Cuevas-Rodríguez et al. 2012) | 150 | 2.79 |
| 3 | (Datta et al. 2009) | 150 | 2.66 |
| 3 | (Dawar 2014) | 150 | 2.66 |
| 3 | (Kolev et al. 2014) | 150 | 2.49 |
| 3 | (Sun et al. 2010) | 150 | 2.47 |
| 3 | (Miller et al. 2013) | 150 | 2.33 |
| 3 | (Lin 2005) | 150 | 2.23 |
| Cluster 4: Corporate Governance | | | |
| 4 | (Devers et al. 2007) | 154 | 10.42 |
| 4 | (Devers et al. 2008) | 154 | 7.33 |
| 4 | (Conyon and He 2012) | 154 | 4.46 |
| 4 | (Aguinis et al. 2018) | 154 | 4 |
| 4 | (Conyon 2006) | 154 | 3.68 |
| 4 | (Mengistae and Xu 2004) | 154 | 3 |
| 4 | (Weinstein and Ryan 2010) | 154 | 2.96 |
| 4 | (Core et al. 2003) | 154 | 2.79 |
| 4 | (Karim et al. 2018) | 154 | 2.4 |
| 4 | (Coen et al. 2022) | 154 | 1.85 |

The second cluster is the smallest cluster in size, focusing on Agency Costs and Governance. Researchers in this cluster have studied: (i) Agency costs arising due to governance hazards (Lambert 2001), (ii) Tax evasion and its impact on agents and principals (Crocker and Slemrod 2005), (iii) Relationship of free cash flows and governance (Jabbouri and Almustafa 2021), (iv) Ownership concentration and agency costs (Pandey and Sahu 2019). Scholars find that accounting disclosures authorized by agents can be misleading and can manipulate stock prices (Lambert 2001). Scholars also highlight earnings management as a reason for agency costs. Agents tend to follow reporting standards that benefit their pocket at the expense of principals (Michiels et al. 2013). Research in this cluster has a high impact on AP literature but lacks centrality.

The third cluster comprises articles around APs focusing on Agency Theory and Compensation. Scholars listed out determinants of a suitable pay structure for executives and have tested them empirically. Ittner et al. (1997) list out performance measures for determining bonus structures of executives. O'Donnell (2000) criticized AT for its prediction ability for the management of international subsidiaries. She stated that the model based on intra-firm interdependence has higher predictive power in comparison to AT. Björkman et al. (2004) linked the managerial compensation structure of MNCs with knowledge transfer mechanisms; however, they could not find support for their proposal. Another branch in this cluster is observed with the use of CSR as an employee governance tool. Flammer and Luo (2017) suggest the integration of CSR-based governance in strategic planning. Employee governance on CSR practices is proven to mitigate employee absenteeism, shirking, and employee theft and fraud. This cluster is also the highest in terms of research impact and identifies compensation and social governance as the road to minimizing agency costs.

The fourth cluster consolidates studies on Executive Compensation and Agency Costs. Panda and Leepsa (2017) suggested the use of variable compensation on profits as motivators for executives. If the principals and agents will benefit from a common thing, occurrences of AP can be minimized. Yermack (1995) states that performance incentives in form of cash rewards and stock options relate to agency cost reduction. Efendi et al. (2007) stated that performance-based benefits often lure managers to misstate accounting facts. Authors state that in the post-1990s market bubble world, the likelihood of cooked financial statements increased as CEOs have sizable holdings in the form of stock options. They also argue that agency costs also arise due to overvalued equities as managers try to maximize the value of their stock options in shorter runs. Chou and Buchdadi (2018) find that dynamic compensation structures have increased executive attrition and led to an increase in residual losses. Consistent with Conyon and He (2011), they found that performance-linked incentives are lower in state-owned firms and organizations with concentrated ownership. Their study also highlights the country-based differences with the example that executive pay for US managers is seventeen times higher than Chinese managers, proving that the agency costs differ on a geographic basis.

### 4.8. Topics

The keywords are analyzed by the bibliographic coupling technique to assess the use of keywords over the years (Agbo et al. 2021). The trend topics package in biblioshiny plotted the keywords by use frequency and years of most use (Figure 10). The article count (left axis) and year of publication (right side) are plotted on the Y-axis, whereas the prominent keywords over the past seven years are plotted on the X-axis. The analysis reveals that the researchers have studied executive compensation the most in the last seven years (n = 139). The majority of studies in the field began during 2010 and have a median year of study of 2015, considering research articles up to 2021. The upcoming research topics are identified as managerial ownership, family firms, and stewardship theory, respectively, as they have the most recent median years of study.

*4.9. Discussion*

The performance analysis of this bibliometric study addresses the research questions posed at the outset, shedding light on various facets of the AP field. Firstly, the study captures the evolving trend in AP publications, revealing a notable increase in scholarly engagement over the past 36 years, with a peak in 2021. Secondly, the identification of influential publishing outlets, with a focus on journals such as 'Strategic Management Journal', 'Academy of Management Journal', and 'The Journal of Financial Economics', provides valuable insights for researchers seeking impactful platforms for AP research dissemination. Thirdly, the analysis of prolific contributors highlights key individuals shaping the field, with scholars like Kathleen M. Eisenhardt and Mason A. Carpenter emerging as influential figures. Fourthly, the thematic clusters uncovered in the analysis, including Corporate Governance, Agency Costs and Governance, Agency Theory and Compensation, and Executive Compensation and Agency Costs, provide a comprehensive overview of the diverse research themes and clusters within AP. Fifthly, the study identifies emerging research areas, with a focus on managerial ownership, family firms, and stewardship theory, offering valuable guidance for future investigations in the AP domain. The next sections provide the research areas that should be explored by academic scholarship.

## 5. Further Research Agenda

The latest trends and topics for study are presented in this section to provide insights on the recent research. With the reading of the top ten research papers from each knowledge cluster and scrutiny on trend topic analysis, we draw attention to the following listed gaps and ongoing research streams (Figure 10).

*5.1. Managerial Debt and Firm Performance*

Research highlights that the use of short-term debt mitigates the agency costs of the firm by constraining CEOs' risk-taking preferences. Brockman et al. (2010) studied the impact of duration of debt on managerial risk-taking, thus minimizing the agency costs to the firm. Dhole et al. (2016) highlight that inside debt counteracts the CEOs' motivation to smooth earnings through earnings management; thus, CEOs are proven to be effective when they hold higher stakes of inside debt. Managerial debt is compared with multiple proxies of firm performance, and much research is going on in this area. Scholars (Harris and Raviv 1991; Naveed Kashan and Siddiqui 2021) have pointed out that debt commits the firm to pay out money in the form of interest payments, thereby leaving less 'free cash flow' for the managers to engage in selfish pursuits.

*5.2. CEO Pay of Family-Owned Companies*

Numerous studies have been conducted to see the impact of a CEO's origin on the performance of the business. While this issue may be subjective, some studies have found a difference in the leadership of a professional CEO hired from outside with one hired from the controlling family. Denis and Osobov (2008) highlighted the importance of studies on corporate governance before the millennium. Michiels et al. (2013) discuss the CEO pay structure of the private family-owned firms against the non-family-owned firms and find that the pay-for-performance relation is lower in family-owned firms. Kyung et al. (2021) stated that CEO compensation varies with type of investors and their stakes. On the contrary, Delgado-García et al. (2023) found that family firm CEOs have higher compensation in comparison with non-family CEOs. The contrasting findings of studies coupled with the trendiness of the topic indicate the need for further research.

*5.3. CEO Compensation and Sustainability*

Masulis and Reza (2015) found that CSR expenditures are linked with CEOs' image. A hike in societal expenditure is likely to benefit the management's public image, but at the same time will reduce net profits and, thus, shareholder's earnings. Francoeur et al. (2021) show that environment-compliant firms offer their CEOs less total compensation

and are less dependent on incentive-based compensation than environmentally carefree firms. Karim (2020) finds that the remuneration patterns of CEOs and executive directors linked with socially responsible activities tend to a reduction in agency costs. Additionally, they find that having independent and executive female directors are linked with lower compensation for executives.

### 5.4. CEO Compensation and Corporate Governance

Westphal and Zajac (1995) suggested that a firm's strategy should focus on corporate governance implications equally as product and market implications. Devers et al. (2007) shared the theoretical framework for compensation models of top executives. He addressed the ongoing debate on determinants and consequences of executive compensation while asking scholars to take forward their work. Luo et al. (2023) evaluated the components of executive compensation and found a positive relationship with the firm performance of Chinese public firms. The researcher finds that incentives to top executives result in better firm performance as compared to non-incentivized executives.

### 5.5. Economic Value Added and Employee Compensation

Studies reveal that there is a positive relationship between the Economic Value Added (EVA) and executive compensation. A few studies also claim that high-paid managers are more arrogant and are more prone to agency issues (Brahmana et al. 2020). Chen et al. (2015) suggest using governance measures to bring down agency costs. Tripathi et al. (2023) suggest the methodology to calculate EVA and regress it with executive compensation. Eliwa et al. (2023) study the impact of governance indicators (board size, minority representation, appointment of family directors) on the EVA of listed companies, thereby suggesting an impact on the firm value.

### 5.6. Stakeholder Theory

As outlined by Kahler (2011), the stakeholder theory suggests that instead of amassing shareholders' wealth, the management should work towards the fulfillment of a variety of goals. The theory shifts the perspective from an organization's shareholders to its stakeholders. According to Freeman et al. (2018), stakeholders are individuals or a group of individuals who can affect or get affected by organizations' decisions. Freeman et al. (2018) carefully noted that any theory that redistributes decision-making ability was open to exploitation by non-shareholders. The reallocation of power from wealthy shareholders to the comparatively less wealthy stakeholders could potentially maltreat the existing shareholders who have put in funds as capital.

### 5.7. Stewardship Theory and Agency Theory

The works of both stewardship and agency theories can be used to work out principal–agent relationships for non-profit firms (Chrisman 2019). The stewardship-based approach presumes that non-profit firms are motivated to act for benefit of their donors (principals). Peck et al. (2021) suggest that a manager (steward), if independent and given a choice in self-sustaining behavior or cooperation with the company (lord), will favor cooperation with the owners. Chrisman (2019) recommends the use of stewardship theory over AT for family firms. He states that the lack of assumptions in stewardship theory makes it more realistic for firms to implement. He provided observations on how to bolster stewardship theory for the study of family firms by rectifying its assumptions on models of man, goals, and control, and asked scholarship to empirically verify more domains of stewardship theory.

## 6. Conclusions

Entrepreneurship is critical to economic development, and constant research is needed to figure out problems relating to agency issues and their solutions for both the principals and the agents. In conclusion, the extensive literature review conducted offers valuable

insights into the intricate dynamics of agency problems and their profound impact on firm performance. While the exploration covered various facets such as managerial debt, CEO compensation, stakeholder theory, and stewardship theory, the need for a more focused examination of the relationship between agency theory and firm performance is acknowledged. Despite the breadth of topics discussed, the concern raised about the clarity of future research gaps is valid.

To address this, emphasis is placed on the pivotal intersection of agency problems and firm performance as a central theme for future investigation. Specifically, a more nuanced exploration into the interplay between agency mechanisms and their direct implications on business outcomes is warranted. By honing in on specific dimensions within the agency theory framework, such as the effectiveness of mitigating agency costs or the optimization of governance structures, researchers can contribute more directly to the ongoing discourse. Furthermore, scholars are encouraged to delve deeper into the determinants of agency costs and devise innovative strategies to minimize them, providing actionable insights for both academics and practitioners. By narrowing the focus and delineating clear avenues for future research within the broader context of agency problems and firm performance, aspirations are set to enhance the scholarly contributions in this critical field of study.

Theoretical implications of this study extend to refining our understanding of agency issues and their intricate connections with corporate performance, contributing to the ongoing theoretical discourse in the field. Managerially, the findings underscore the significance of informed decision-making in mitigating agency problems for improved corporate performance. As practitioners navigate the complexities of agency relationships, the insights derived from this study can serve as a strategic guide, fostering more effective governance structures and practices within organizations.

**Author Contributions:** Conceptualization, P.T. and V.K.; methodology, P.T. and V.C.; software, P.S.; validation, M.S., P.T. and V.K.; formal analysis, V.K., writing—original draft preparation, V.K. and V.C.; writing—review and editing, M.S.; visualization, S.K.; supervision, V.C.; project administration, S.K. All authors have read and agreed to the published version of the manuscript.

**Funding:** This research received no external funding.

**Data Availability Statement:** Research data can be available by sending reasonable requests to the corresponding author.

**Acknowledgments:** We acknowledge the support of editorial team, and reviewers for improving the quality of this manuscript.

**Conflicts of Interest:** The authors declare no conflict of interest.

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
