# Peer review of "Examining the Impact of Agency Issues on Corporate Performance: A Bibliometric Analysis"

_jrfm, doi:10.3390/jrfm16120497_

Round 1

Reviewer 1 Report

Comments and Suggestions for Authors

The article is well-presented and interesting to read in relation to the topic of the literature review on agency theory and corporate performance. However, I detect some issues that need to be addressed by the authors to improve the quality of the current work. Plese refer to the attachment.

Author Response

Dear Editorial team, and anonymous reviewers, we are thankful for your support in enhancing the quality of this manuscript. The comments of all the reviewers are aggregated in this sheet, along with the authors’ responses to each of the comments.

Reviewer 1 Comment 1 (R1C1):

  1. The article is well-presented and interesting to read in relation to the topic of the literature review on agency theory and corporate performance. However, I detect some issues that need to be addressed by the authors to improve the quality of the current work.
  2. I found some citations that are not listed in the reference list. A citation is mentioned several times, such as Panda and Leepsa (2017), but does not appear in the reference. Please be aware of this issue. Please use the reference manager app to solve this issue. Here is the list of citations:

  1. Panda and Leepsa (2017), lines 42, 102, and 364
  2. Trinarningsih et al. (2021), line 63
  3. Brahmana et al. (2021), line 130
  4. Kontesa et al. (2021), line 446
  5. Dounthu et al. (2021), line 166, but in the reference list it is stated Donthu et al. (2021); in line 244 it is stated Donthu et al. (2021).

Are these referring to the same article? Please be consistent in style in writing the title in the reference list. Mitnick (1982); the title is written in capital letters.

Response to R1C1:

We thank you for your comments. All the citations are now thoroughly checked and updated using Mendeley Reference Manager. All the missing citations are corrected with the use of software. The spelling and formatting errors are also rectified, and manually checked for above issues.

Reviewer 1 Comment 2 (R1C2):

  1. In relation to Table 5, I suggest that the authors explain what they mean about centrality and impact in the table and elaborate how to assess them specifically the interconnectedness with the discussed theme. Probably it is better that the authors add one more column in Table 5 which mention the theme in the cluster. e.g. Corporate Governance, Agency Cost and Governance, etc.

Response to R1C2:

We thank you for your comments. The inputs regarding centrality and impact have been added in section 4.7. The cluster themes have also been added to the table 5 per your suggestion.

Reviewer 2 Report

Comments and Suggestions for Authors

The title indicates that there will be a review of the relationship between firm performance and agency theory but the review discuss many topics with no clear gaps for future research. The contribution is minimum.

Author Response

Dear Editorial team, and anonymous reviewers, we are thankful for your support in enhancing the quality of this manuscript. The comments of all the reviewers are aggregated in this sheet, along with the authors’ responses to each of the comments.

Reviewer 2 Comment 1 (R2C1):

  1. The title indicates that there will be a review of the relationship between firm performance and agency theory but the review discuss many topics with no clear gaps for future research. The contribution is minimum.

Response to R2C1:

We thank you for your comments. We have modified our concluding paragraphs for this issue. Th e paragraph now provides the link of different identified clusters with the agency problems and its impact on the firm performances. The clear gaps are also summarized for future researches to explore upon.

Reviewer 3 Report

Comments and Suggestions for Authors

Thank you for the opportunity to read this article.

The review of the literature is quite complex and well-developed.

The methodology section should be supplemented with information about the software used to develop this research.

The conclusion section should focus more on the theoretical and managerial implications.

The author(s) can introduce, separately from the conclusion, a discussion section based on a critical synthesis approach or rename section 4 as ”Results and discussions”. The discussion should connect to the introduction through the research questions and the literature reviewed. The author(s) should reiterate the research problem, stating the major findings.

Comments on the Quality of English Language

The article is written in understandable English with concise and clear sentences.

Author Response

Reviewer 3 Comment 1 (R3C1):

  1. The methodology section should be supplemented with information about the software used to develop this research. The conclusion section should focus more on the theoretical and managerial implications.

Response to R3C1:

We thank you for your comments. We have added the software details in the methodology section. Additionally, we have also added the focus on theoretical and managerial implications of this research in the conclusions section.

Reviewer 3 Comment 2 (R3C2):

  1. The author(s) can introduce, separately from the conclusion, a discussion section based on a critical synthesis approach or rename section 4 as ”Results and discussions”. The discussion should connect to the introduction through the research questions and the literature reviewed. The author(s) should reiterate the research problem, stating the major findings.

Response to R3C2:

The fourth section is renamed to “Results and Discussion”. The sub-section 4.9 is added as discussion, summarizing the findings of performance analysis and achievement of research objectives. The answers to research questions are also framed in the sub-section, thus, improving the readability of the manuscript. We thank the reviewers for their suggestions on improving this section of the manuscript.

Reviewer 4 Report

Comments and Suggestions for Authors

Title: Examining the Impact of Agency Issues on Corporate Performance: A Bibliometric Analysis for JRFM

The article analyzes 740 research articles on agency problems indexed in Scopus using bibliometric techniques. It examines publishing trends, influential journals, prolific authors, institutional collaborations, prominent countries, research themes, and future research agenda. Four major research themes identified through bibliographic coupling are corporate governance, CEO compensation, agency theory, and executive compensation & agency costs.

The article provides a comprehensive overview of research on agency problems using bibliometric analysis and identifies promising areas for future scholarship. It highlights the importance of this topic and the need to further explore solutions to mitigate agency costs.

I found the paper well written and the study was comprehensive. I will encourage the authors to consider the following suggestions:

1.       The abstract should specifically state the value addition and novel contribution of this bibliometric study to the literature on agency problems. This will make the purpose clear upfront.

2.       The introduction should provide more background and context about the emergence of agency theory and problems before diving into the bibliometric study. Tracing the historical origins will help readers. The introduction should clearly highlight and summarize the specific gaps in literature that this study aims to address. This will clarify the purpose and need for the study. Strengthen the framing by clearly articulating the need, purpose, novel contribution, and implications early on. State the specific gaps in literature on agency problems that this study addresses. Highlight value addition.

3.       Both the introduction and abstract should briefly explain the value of bibliometric analysis and its utility for examining the state of research in a field like agency problems. This will justify the chosen methodology.

4.       Expand the literature review to provide more background and trace the evolution of agency theory research over time. Add a section to discuss the origins of agency theory, key developments chronologically (e.g. Berle and Means' work in 1930s, Eisenhardt's theories in 1980s etc.) and theoretical progress in defining agency issues.

5.       Widen the literature search beyond Scopus to include other relevant scholarly databases. Search Web of Science, ABI/Inform, EBSCOhost and other databases relevant to business, management, accounting, and finance. At the very least do a check to see if any area of AP is being systematically missed.

6.       Apply filters by methodology, impact, citations etc. and explain the inclusion criteria clearly to get high quality articles. Add filters to limit by methodology, exclude editorials/books, delineate by citations. State exact criteria for article inclusion/exclusion. Self-citations should be excluded - The citation analysis may be inflated by author self-citations which are not filtered out. This skews impact.

7.       Conduct an in-depth content analysis of top cited articles to reveal gaps and guide future research. Review the research questions, theories, variables, findings and limitations of highly cited articles in each theme.

8.       Improve statistical robustness by using fractional authorship, normalizing citations, and testing for significance. Use fractional counting for authors, field-normalize citations, add statistical tests for comparing author/country productivity.

Comments on the Quality of English Language

      Overall, the paper is well written. Writing may be improved by considering the following suggestions: Refine the writing style through active voice, variety, concision, grammar fixes and logical flow between sections. Change passive phrases to active voice. Vary sentence structures. Eliminate redundancies and wordiness.

Author Response

Dear Editorial team, and anonymous reviewers, we are thankful for your support in enhancing the quality of this manuscript. The comments of all the reviewers are aggregated in this sheet, along with the authors’ responses to each of the comments.

Reviewer 4 Comment 1 (R4C1):

  1. The abstract should specifically state the value addition and novel contribution of this bibliometric study to the literature on agency problems. This will make the purpose clear upfront.
  2. The introduction should provide more background and context about the emergence of agency theory and problems before diving into the bibliometric study. Tracing the historical origins will help readers. The introduction should clearly highlight and summarize the specific gaps in literature that this study aims to address. This will clarify the purpose and need for the study. Strengthen the framing by clearly articulating the need, purpose, novel contribution, and implications early on. State the specific gaps in literature on agency problems that this study addresses. Highlight value addition.
  3. Both the introduction and abstract should briefly explain the value of bibliometric analysis and its utility for examining the state of research in a field like agency problems. This will justify the chosen methodology.

Response to R4C1:

We thank you for your comments. The abstract has been modified to reflect the value addition and novel contribution of this manuscript. The introduction section has also been extended to include the origin of Agency theory addressing the agency problems and associated costs. The expanded evolution has been discussed in the second section. A paragraph is added in the methodology section justifying the suitability and use of bibliometric analysis for examining the state of research in AP domain, justifying the chosen methodology.

Reviewer 4 Comment 2 (R4C2):

  1. Expand the literature review to provide more background and trace the evolution of agency theory research over time. Add a section to discuss the origins of agency theory, key developments chronologically (e.g. Berle and Means' work in 1930s, Eisenhardt's theories in 1980s etc.) and theoretical progress in defining agency issues.
  2. Widen the literature search beyond Scopus to include other relevant scholarly databases. Search Web of Science, ABI/Inform, EBSCOhost and other databases relevant to business, management, accounting, and finance. At the very least do a check to see if any area of AP is being systematically missed.

Response to R4C2:

We thank you for your comments. The roots and evolution of agency theory, especially the works of Berle and Means in 1932, and Eisenhardt’s theories of 1985, 1989, have now been added to the theoretical evolution section of the manuscript. We, sincerely, thank you for helping us improve the quality of this manuscript. This manuscript could not summarize the articles from other databases due to lack of access to the prestigious databases, and the technical knowhow on integrating the review corpus. We would consider this as a way forward and explore ways to do so. The areas are, however, checked, and we assure you that majority studies in this domain are covered, and have been summarized in this review article.

Reviewer 4 Comment 3 (R4C3):

  1. Apply filters by methodology, impact, citations etc. and explain the inclusion criteria clearly to get high quality articles. Add filters to limit by methodology, exclude editorials/books, delineate by citations. State exact criteria for article inclusion/exclusion. Self-citations should be excluded - The citation analysis may be inflated by author self-citations which are not filtered out. This skews impact.
  2. Conduct an in-depth content analysis of top cited articles to reveal gaps and guide future research. Review the research questions, theories, variables, findings and limitations of highly cited articles in each theme.
  3. Improve statistical robustness by using fractional authorship, normalizing citations, and testing for significance. Use fractional counting for authors, field-normalize citations, add statistical tests for comparing author/country productivity.

Response to R4C3:

We thank you for your comments directed toward improving the publishability of this manuscript. We have modified the search strategy section to reflect the process through an inclusion-exclusion lens. The editorials, books, etc. have been excluded in the scholarly filtration stage to maintain the quality of review corpus. The exact criteria followed is presented in the section. We lack knowledge on how to check for self-citations for all the articles in review corpus (740), therefore using the local and global citations to rank the published articles, as used by other similar studies. An analysis of top-cited papers is summarized in the themes, and knowledge clusters sections. The same is not presented in the results and discussion section to avoid wordiness and duplicity. The in-depth review of research questions, theories, and variables could not be performed due to lack of resources. This study uses the tools and techniques provided in the bibliometrix package of R and used through reference of other similar studies. We value your suggestions, and will build our competence on use of these advanced metrices for statistical robustness.

Reviewer 4 Comment 4 (R4C4):

  1. Overall, the paper is well written. Writing may be improved by considering the following suggestions: Refine the writing style through active voice, variety, concision, grammar fixes and logical flow between sections. Change passive phrases to active voice. Vary sentence structures. Eliminate redundancies and wordiness.

Response to R4C3:

The revised version of this paper has been re-read and verified by peers for improvement of writing. The grammar check software, Grammarly, has also been used to rule out punctuation and grammar errors. The redundancies and wordiness have been checked. We thank you for your suggestions, they have helped us in improving the readability of this manuscript.

Round 2

Reviewer 2 Report

Comments and Suggestions for Authors

The contribution of this paper is still very minimum. Authors should consider new directions in agency theory research to make their paper valuable. Such directions include Artificial Intelligence and Blockchain and their relation with agency theory. 

Reviewer 4 Report

Comments and Suggestions for Authors

Thank you for incorporating the comments and suggestions.

Wishing you the best for next stage!